# Cytotoxic and Phytotoxic Activities of Native Brazilian Forest Gabiroba (*Campomanesia xanthocarpa* Berg.), Fruits, and Flour against Shrimp (*Artemia salina* L.) and Lettuce (*Lactuca sativa* L.)

**DOI:** 10.3390/foods13010123

**Published:** 2023-12-29

**Authors:** Aiane Benevide Sereno, Carla Dayane Pinto, Luciana Gibbert, Marina Talamini Piltz de Andrade, Michelli Aparecida Bertolazo da Silva, Schaina Andriela Pontarollo Etgeton, Obdulio Gomes Miguel, Josiane de Fátima Gaspari Dias, Claudia Carneiro Hecke Krüger, Iara José de Messias Reason

**Affiliations:** 1Graduate Program in Internal Medicine, and Health Sciences, Federal University of Paraná (UFPR), R. Padre Camargo, 280, Curitiba 80.069-900, Paraná, Brazil; nutri.carladay@gmail.com (C.D.P.); michelli.bertolazo@yahoo.com.br (M.A.B.d.S.); iaramessias@yahoo.com.br (I.J.d.M.R.); 2Graduate Program in Pharmaceutical Sciences, Federal University of Paraná (UFPR), Av. Lothário Meissner, 632, Curitiba 80.210-170, Paraná, Brazil; luci.gibbert@gmail.com (L.G.); marinapiltz@yahoo.com.br (M.T.P.d.A.); obdulio@ufpr.br (O.G.M.);; 3Graduate Program in Food, and Nutrition, Federal University of Paraná (UFPR), Av. Lothário Meissner, 632, Curitiba 80.210-170, Paraná, Brazil; schainaetgeton13@gmail.com; 4Department of Nutrition, Federal University of Paraná (UFPR), Av. Lothário Meissner, 632, Curitiba 80.210-170, Paraná, Brazil; 5Department of Medical Pathology, Clinical Hospital, Federal University of Paraná (UFPR), R. Padre Camargo, 280, Curitiba 80.069-900, Paraná, Brazil

**Keywords:** nutritional composition, cytotoxicity, phytotoxicity, stabilized gabiroba flour, *Campomanesia xanthocarpa* Berg

## Abstract

Gabiroba, a native fruit in Brazil’s Atlantic Forest region, has significant nutritional and therapeutic properties. However, due to its seasonality, consumption by the population is limited. Thus, the development of gabiroba byproducts would add significant value to the food and therapeutic industries. Therefore, it is essential to study and support the lack of toxicity of gabiroba fruit extracts. In the present study, physicochemical analyses of fresh fruits (GF) and dehydrated whole gabiroba flour (WGF) and preliminary toxicity analyses of WGF were performed. The toxicity results showed a microcrustacean LC50 of >1000 mg/mL when exposed to WGF extracts at various concentrations (10–1000 μg/mL; *p* = 0.062) using the *Artemia salina* method, with no evidence observed of proliferative activity or toxic metabolic compounds in the WGF extract. The phytotoxicity of WGF using *Lactuca sativa* L. allowed germination and root growth at various concentrations of WGF extract, with the lowest (100 μg/mL) and highest (1000 μg/mL) concentrations exhibiting 98.3% and 100% seed germination, respectively. In conclusion, these results indicate that the WGF preparation preserved the nutritional and antioxidant potential of gabiroba fruits and that WGF is safe for use as a raw material in the food industry and for therapeutic purposes.

## 1. Introduction

The cultivation of native fruits is important in terms of economic, social, and sustainable aspects worldwide, with Brazil playing a vital role in this context, given its vast biodiversity [1,2,3]. Among the primary native fruits found in the Brazilian Atlantic Forest are the abundant species of the Myrtaceae family, with relevance to the diets of several regional cultures [3].

*Camponesia xantocarpa* Berg., commonly known as gabiroba, guabiroba, or guaviroba, belongs to the Myrtaceae family. Although it is found in several regions of Brazil, it is not considered a popular food for routine consumption, highlighting the importance of introducing new added value, specifications, and applications [3]. Gabiroba (GF) has a high content of dietary fiber (4.1–9.8%), vitamin C (826.26 mg/100 g), carotenoids (α-carotene, 4.8 µg/g; β-carotene, 5.4 µg/g; violaxanthin, 2.84–4.5 µg/g; β-cryptoxanthin, 5.8 µg/g; and lutein, 4.5–14.92 µg/g), and total phenolic compounds (19.59–131.9 µg/g^−1^), thus promoting its antioxidant capacity [4,5]. Its fruits are consumed fresh in the form of pulp and are used as raw materials for preparing drinks, ice cream, jellies, and flour [6,7]. 

Several studies have reported the health benefits of consuming GF and other fractions of the Gabirobeira tree. In vitro and animal experiments have shown that the traditionally used leaves exhibit hypotensive activity [8,9] and a hypocholesterolemic effect in humans in double-blind randomized clinical trials (*n* = 200) [10]. The seeds demonstrated hypoglycemic potential in an oral glucose tolerance test in an animal study [11] and antimicrobial effects in vitro [12]. The pulp exhibited prebiotic activity in gastrointestinal simulations [13], and the fruit extracts showed antiproliferative effects in cancer cell lines [14] in in vitro studies.

Considering the genetic and morphological variability of GF and the influence of the soil of each region on phytochemical substances [15,16], it is essential to investigate and characterize the fruit grown in the Atlantic Forest of Southern Brazil and its by-products stabilized for consumption (whole flour).

Furthermore, considering the use of GF either as a food or raw material for medicinal purposes, it is fundamental to investigate its toxicity. The lethality test against *A. salina* L. is widely used to evaluate the toxicity levels of several substances, including extracts and phytochemicals isolated from natural products [17,18]. In addition to testing the microcrustacean *A. salina*, the phytotoxicity of plant extracts in bioassays can be identified using *L. sativa* (lettuce) via the chemical control method from the inhibitory interference of seed germination and seedling growth upon exposure to the test plant [19].

Allelopathy, as defined by Molisch [20], refers to the chemical interactions of plants with stimulatory and inhibitory effects. Knowledge of the allelopathic activity (phytotoxicity) of native cultivars is still lacking in Brazil, considering the territorial extension and biodiversity of the country [21]. This low-cost and highly sensitive method is used to assess the toxicity of a specific species on the development and germination of other species. Investigations on the allelopathic activity of gabiroba are still limited. Therefore, in the present study, the cytotoxicity and acute phytotoxicity of gabiroba fruit and GF have been investigated, as well as their physical-chemical composition and antioxidant activity for comparative purposes with the nutritional stability of dehydrated whole gabiroba flour (WGF).

## 2. Materials and Methods

### 2.1. Raw Material

The collection of native fruits took place in the Brazilian Atlantic Forest (25°52′08.1″ S/51°33′03.9″ W) in November 2019. After collection, the fruits were transported in isothermal boxes (18 ± 2 °C) and selected by the degree of ripeness. Fruits were considered ripe when more than 80% of their peel had an orange hue (D scale) [16] (Figure 1). This study was registered in the National System for Genetic Heritage Management (SISGEN): A803326O. The material was botanically identified at the Curitiba Municipal Botanical Museum, and an exsiccate was deposited in the herbarium (MBM 36735).

The selected samples were immersed in a chlorinated solution (200 mg/L sodium hypochlorite) for 15 min, rinsed in running water, and dried at room temperature. They were then separated for physical and physicochemical analysis and stored under refrigeration at 3.3 ± 0.5 °C. Part of the fruits was frozen (−20 °C ± 2) and freeze-dried (Cperon Freezer Dryer −55 °C, São Paulo, Brazil) at −58.0 ± 2.0 °C and 50.0 ± 10.0 µHg for 72 h for antioxidant capacity evaluation. Portions of 20 g of freeze-dried fruits were vacuum packed (Jumbo Plus, Selovac, São Paulo, Brazil) in polyethylene bags and stored at −18.0 ± 0.5 °C in a conventional freezer until the process of obtaining extracts.

In order to compare the results and the physicochemical stability of the fruits with their dehydrated flour, whole gabiroba flour (with seeds and peels) (Figure 2) was purchased from the same biome (25°51′58.9″ S/49°08′49.8″ W), processed via dehydration, and without the use of additives, preservatives, or allergens (Quinta das Cerejeiras Alimentos Inteligentes LTDA, Tijucas do Sul, Brazil). 

According to the manufacturer’s information, the flour was made by choosing healthy fruits, free from stains and dirt. The selection was carried out manually with subsequent washing in running water. The selected and washed fruits were sliced and/or cut to facilitate the dehydration process. Afterward, drying took place, and at this stage, the water was removed from the fruits until a humidity of less than 10% was reached. Periodically moving the trays reduced the drying time and guaranteed the homogeneity of the final product (the temperature was approximately 60 °C until the end of the process). Heat transfer in the oven-type dryer occurs via the forced hot air circulation. The chamber receives the trays with the fruits, and the heated air is ventilated from an attached compartment where the heating system is located. The heated air is transferred to the drying chamber through an air recirculation system between the trays (shelves). In the grinding stage, the dehydrated fruits are subjected to grinding (shredding) in a hammer mill. The mill uses a system of sieves that allows the average size of the powder particles to be standardized. When sieving, the manufacturer uses a special sieve, with a stainless-steel sieve, whose mesh is specific to this product, which allows the average size of the powder particles to be standardized, presenting a homogeneous granulometry. Commercial packaging ensured that the dehydrated product was protected against the action of humidity, air, and light. 

Whole gabiroba flour was coded in this study as WGF. The fruit, expressed in its edible portion (pulp, peels, and seeds) was coded as GF.

#### 2.1.1. Process of Elaboration of WGF Extract

Initially, the aqueous extracts of the fruits and the WGF were obtained according to the methodology proposed by Bursal and Gülçin [22] and adapted with methanol (0.1%) for the analysis of the phytotoxicity against *Lactuca sativa* L. For this, the GF, and the WGF were homogenized under ultrasound using distilled water as the solvent (15 min). Afterward, the dispersion was kept for 24 h under magnetic stirring at room temperature (Fisatom 753, Curitiba, Brazil). 

Then, the ethanolic extract was prepared according to the methodology proposed by Maria do Socorro et al. [23]. The WGF and lyophilized fruits were dissolved in ethanol:water (50:50) (*v*/*v*) and centrifuged at 2000 rpm for 15 min (Fanen 280R, São Paulo, Brazil). Afterward, a new extraction was performed with acetone:water (70:30) (*v*/*v*). The extraction residual was removed in a rotary evaporator (USC-1400 Unique, São Paulo, Brazil). After the preparation of the extract, dilutions were performed to obtain concentrations of 100, 250, 500, 750, and 1000 µg/mL. These dilutions were used for the cytotoxicity analyses against *Artemia salina*. During all procedures, the extracts were handled and stored in the absence of light, avoiding the degradation of the photosensitive compounds [24].

#### 2.1.2. Physical Chemical Compositions of GF and WGF

The moisture content (AOAC 925.09) and fixed mineral residue (AOAC 923.03) were evaluated using gravimetric methods at atmospheric pressure, according to the Association of Official Analytical Chemists [25]. The protein content was evaluated using the Kjeldahl technique, with a nitrogen correction factor of 5.75 (AOAC 920.152), and the lipids using the Soxhlet method with petroleum ether (AOAC 930.09) [26]. Dietary fiber was obtained via the enzymatic–gravimetric process in a dietary fiber analyzer (CSF6—Velp Scientífica, Usmate, Italy) (AOAC—985.29) [25]. The content of total soluble solids (TSSs) and titratable acidity (TA) obtained using the concentration of 0.1 mol/L of sodium hydroxide (NaOH) with a correction factor for hydrochloric acid of 1.098 (AOAC, 942.15) were analyzed, and the results are expressed in grams of organic acid (g/100 g) [26]. The total carbohydrate content was calculated by the difference of 100%, and the sum of the percentages of moisture, ashes, proteins, lipids, and total fibers, according to Millar [27]. The total energy value (TEV) was obtained using the formula proposed by 100.

The ferric reducing antioxidant power (FRAP) assay was performed using the method proposed by Arnous et al. [28] with adaptations. The extract (0.1 mL) and an aliquot of 0.1 mL of FeCl_3_ (3 mM in 5 mM citric acid) were homogenized and incubated for 30 min in a water bath at 37 °C. Subsequently, 0.9 mL of TPTZ solution at 1 mM in 50 Mm hydrochloric acid (HCL) was added, this mixture was vortexed, and after 10 min, it was possible to obtain the absorbance reading at 620 nm with a Hewlett-Packard spectrophotometer. The Packard model HP of Atwater and Bryant [29,30] was used, where TEV = (% total protein × 4) + (% carbohydrates × 4) + (% fat × 9), and the results are expressed in kcal per 100 g. The physicochemical analyses were performed in triplicate.

#### 2.1.3. Antioxidant Activity of GF and WGF

To evaluate the scavenging of free radicals by 2,2-diphenyl-1-picrylhydrazyl (DPPH), the methodology described by Brand Williams, Cuvelier, and Berset [31] was used. Considering the final absorbance at 544 nm (t30 min), its resulting antioxidant capacity was obtained via the standard curve with Trolox solution (TEAC) at different concentrations using the following calculation: % inhibition (μmol) = [1 − (absorbance sample (t at 30 min)/absorbance control (t at 0 min))] × 8452A (Cheadle Heath, Stockport, Cheshire, United Kingdom).

Finally, the oxygen radical absorption capacity (ORAC) was investigated. Briefly, 20 μL of the extracts was added to 120 μL of fluorescein solution (61.2 nM) and incubated for 10 min at 37 °C. Then, 60 μL of dihydrochloride solution (AAPH) (19 mM) was added to the microplates, starting the reaction. The fluorescence amplitude was kinetically evaluated every minute (excitation: 485/20 nm and emission: 528/20 nm) until the fluorescence value reached a percentage lower than or equal to 0.5% of the initial fluorescence. The antioxidant capacity was expressed in μmol equivalent of Trolox/100 g [32]. 

#### 2.1.4. Acute In Vitro Cytotoxicity of WGF Extract

Eggs of brine shrimp microcrustaceans (*Artemia salina* L.) were purchased commercially to carry out the preliminary toxicity test according to Meyer et al. [33]. An amount of 200 mg of eggs was hatched in saline water solution (200 mg/400 mL) in a controlled environment with a pH of 8–9, temperature of 27 ± 3 °C, lighting at 20 W, agitation, and constant aeration for 48 h. After hatching, ten nauplii were transferred to test tubes. Then, gradual concentrations of lyophilized aqueous extracts of whole gabiroba flour were added—10, 100, 250, 500, 750, and 1000 μg/mL, in quadruplicate—and 2.5 mL of saline solution was added. The negative controls were kept in saline water and methanol (2:1), and the positive control was kept in quinidine sulfate.

After 24 h, the count of mobile nauplii (survivors) was performed, and the sample was considered non-toxic when concentrations greater than 1000 μg/mL did not cause >50% death of the larvae. The results are expressed in lethal concentration (LC_50_) values to compare the cytotoxicity of the extracts, where % Mortality = (test − negative control mean number of dead) × 100/positive control mean number of dead.

#### 2.1.5. Phytotoxicity Bioassay of WGF Extract

Phytotoxicity occurred with germination, and allelopathic growth with seeds of *Lactuca sativa* L. (lettuce), using the methodology adapted from Chon et al. [34], Days et al. [35], and Macías et al. [36]. The gabiroba aqueous extracts with 0.1% methanol at concentrations of 100, 250, 500, 750, and 1000 µg/mL were transferred to Petri dishes with filter paper (Whatman™, n° 6; Maidstone, UK), previously autoclaved at 120 °C for 20 min. All extracts with the negative control (methanol) remained for 24 h under laminar flow at room temperature for solvent evaporation (methanol). In the positive control treatment, the solution was water with 0.1% methanol, and in the negative control, it was only methanol. After 24 h, 20 lettuce seeds were sown on each plate, in triplicate for each solution. The plates were placed in a germination chamber (BOD) under conditions of constant light (160 W), a controlled temperature at 25 °C, and a relative humidity at 80%.

##### Germination Test of WGF Extract

Seeds were considered germinated when 2 mm protrusion and geotropic radicle curvature were selected as seed germination criteria. Seeds that showed false germination via soaking were excluded from the count. Germination was monitored by daily counting of the number of germinated seeds during seven days of exposure to the control and treatments [37]. The indices used to evaluate germination and seedling growth with the germination speed index (GSI) were calculated as described by Maguire [38].

##### Growth Test of WGF on Seedlings

After seven days of germination, the radicle and hypocotyl lengths of the seedlings in each plate (*n* = 20 for each experimental group) were measured on graph paper (mm) [37,39]. 

#### 2.1.6. Statistical Analysis

The analyses of physical-chemical composition and antioxidant capacity are expressed as means ± standard deviation, analyzed using GraphPad Prism 6 software (Graphpad Software, Inc., San Diego, CA, USA). Differences between means followed the ANOVA analysis of variance, followed by the Tukey test (*p* < 0.01). For the statistical analysis of cytotoxicity against *Artemia salina*, the probit test with linear regression using the SPSS Statistics software 29.0.19 was used, with a confidence interval of 95%. The results of allelopathic activity were analyzed using the Scott-Knott test, using the Sisvar 5.7 program [40]. 

## 3. Results and Discussion

### 3.1. Physical Characteristics of GF

The losses were minimal when the fruit was cut, and the yield of the edible fraction after removing the peduncle was 98.84%. During the physical characterization process, we noticed that the peduncles were small and almost non-existent in some fruits. This characteristic is easily observed in ripe fruits, allowing for an essential yield for agro-industrial applications. 

Regarding physical characteristics, the fruits had a minimum length of 8.52 ± 0.08 cm and a maximum length of 24.94 ± 0.06 cm, with a density of 2.11 ± 0.37 g/cm^3^.

The observed average weight of the GF was 8.02 ± 2.32 g, which is similar to that of the fruits collected in the state of Rio Grande do Sul, which ranged from 2.47 to 9.37 ± 1.85 g, with an average weight of 5.09 g. Small differences observed in fruit weight resulted from phenotypic characteristics influenced by the soil of each region and plant cultivation [35]. 

### 3.2. Physicochemical Characterization of the GF and WGF

The physicochemical compositions of the edible parts of the GF and WGF are shown in Table 1.

GF presents a high moisture content (81.18 ± 0.39%), similar to that of other fruits of the same family, such as guava cultivars (*Psidium guajava* L.; 82.9 ± 0.50% to 84.3 ± 1.20%) [36], uvaia (*Eugenia pyriformis*; 91.75 ± 0.13%) [37], and jambolan (*Syzygium cumini*; 84.74%) [39]. Because of its high moisture content, GF has a short postharvest life period in its fresh form, requiring conservation techniques for application in the food industry [38], such as the creation of flours that can retain the microbial stability of the product owing to the reduction in water content [40]. Therefore, WGF met this objective by presenting a lower moisture content than GF (*p* > 0.005). 

The results also show a fixed mineral residue (ashes) amount of 0.29 ± 0.07 g/100 g in GF and 3.03 ± 0.15 g/100 g in WGF. Etgeton et al. [41] determined the nutritional composition of GF powder and showed approximate values for the ash content (2.37 ± 0.02). However, it contains a lower amount of protein (2.84 ± 0.03) compared with WGF. As expected, higher protein, lipid, and dietary fiber contents were observed in the WGF compared with those in the GF. Notably, WGF is highly concentrated, equivalent to 8 kg of fresh product to 1 kg of dry fruit. Interestingly, fruit residues such as WGF were shown to contain essential nutrients, with a higher proportion in relation to edible fresh fruit [42,43,44,45]. In addition, the high protein content shown in WGF corresponds to 9.5% of the recommended daily intake (RDI) for adults [46]. Considering that the primary protein sources are of animal origin [47], WGF has emerged as an excellent protein source for plant-based diets [48,49]. 

According to the Technical Regulation on Complementary Nutritional Information, GF has a low lipid content [50]. Foods with a low total fat content must contain values of <3 g/100 g, with GF conforming to these parameters. This value in lipid content is similar to that observed in other gabiroba species from different regions of Brazil (0.1–3.7 g/100 g) and that which is expected in fruits [51]. Furthermore, the WGF lipid content is analogous to that observed in gabiroba waste flour (11.26 ± 0.08 g/100 g) from the pulp industry at the Ouro Verde Reserve, the municipality of Miranda, Mato Grosso do Sul, Brazil [52]. 

Among macronutrients, gabiroba, similar to most fruits, contains a predominance of total carbohydrates. GF has a carbohydrate content similar to that of cultivars from other regions of Brazil, such as those from the Assis State Forest, São Paulo, Brazil (8.9 ± 0.1 g/100 g) [53], and that from Pelotas, in the municipality of Rio Grande do Sul (15.68 ± 0.23 g/100 g), which is the most similar to the carbohydrate content of GF [54]. Although the WGF carbohydrate content reached approximately 54.73% of the 130 g minimum daily recommendation of the American Dietetic Association (ADA) [55], it can be increased as an energy and nutritional percentage in bread formulations with whole flours. 

Both GF and WGF have a considerable dietary fiber content (16.89 ± 3.39 g/100 g) compared with foods rich in fiber present in the daily diet of Brazilians, such as oats in fine flakes (11.3 g/100 g) and whole rye flour (15.5 g/100 g) [56]. Previous studies have shown a positive impact of dietary fiber, particularly soluble fiber, on blood glucose levels. These associations are consistent with those of a study conducted by Biavatti et al. [57], which reported a 15% reduction in plasma glucose levels in rats with induced obesity treated with gabiroba extract. 

Generally, fruits are low-energy-density foods. Thus, GF has a low total energy value (TEV). Considering a portion of approximately 12–13 fruits, GF would provide 4.25% of the RDI (DV%) for adults based on a 2000 kcal diet [58]. In accordance, WGF has a higher TEV than GF and is approximately 105 kcal more than that of wheat bran (277 g/100 g) [56]. It is assumed that not all whole-grain products are necessarily less caloric than their refined equivalents [59]. They provide a higher protein and dietary fiber content, in addition to a lower saturated fat content in whole fruit flour byproducts [60,61].

### 3.3. Antioxidant Activities of WGF and GF

The 2,2-diphenyl-1-picrylhydrazyl (DPPH) assay showed a higher antioxidant activity for WGF (681.72 ± 12.35 µmol TEAC.100 g^−1^) (Table 2) compared with GF (380.64 ± 43.62 µmol TEAC.100 g^−1^) (*p* = 0.0003). This suggests that the fruit dehydration process used to obtain WGF did not cause a loss of antioxidant potential but instead increased this activity. In addition, an oxygen radical absorbance capacity (ORAC) value of 2824.00 ± 11.79 µmol TEAC.100 g^−1^ was observed.

GF also exhibited good antioxidant activity, which was primarily determined using the DPPH radical-reducing method and ORAC. Sereno et al. [60] recently reported that the antioxidant activity of the bark of cocona grown in the Atlantic Forest on the coast of Paraná exhibited antioxidant capacity similar to that of gabiroba in terms of reducing DPP and FRAP radicals (360.49 ± 13.13 µmol TEAC.100 g^−1^ and 137.09 ± 1.13 µmol TEAC.100 g^−1^, respectively). When comparing gabiroba with other fruits from the same family (Myrtaceae), it shows better antioxidant activity than red jambo (*Syzygium malaccense* L.O), grown in southern Brazil (DPPH = 160.73 ± 34.31; ORAC = 372.42 ± 9.50 µmol TEAC.100 g^−1^) [62]. All studies used similar methods to assess antioxidant activity. 

### 3.4. Acute In Vitro Cytotoxicity of WGF Extract

The viability of microcrustaceans for WGF extract was LC_50_ ≥ 1000 mg/mL. The WGF extracts in minimum (10 µg/mL) and maximum (1000 µg/mL) concentrations demonstrated an absence of lethality (*p* = 0.062) and a comparatively lower mortality rate of *A. salina* than the positive control (*p* = 0.898) (Figure 3).

For the extract of a biological species to be considered toxic, its concentration must have a median lethal concentration (CL_50_) of <1000 μg/mL after 24 h of application [29]. Amarante et al. [63] presented another classification system: low toxicity (CL_50_ > 500 µg/mL), moderate toxicity (CL_50_ ranging from 100 to 500 µg/mL), and high toxicity (CL_50_ < 100 µg/mL). Thus, the WGF extract indicates that the plant species is safe for use, enabling its dietary application and supporting further research in experimental animal trials.

Considering this, in this study, WGF was the dehydrated edible fruit, supporting the absence of cytotoxicity of its extract, which becomes relevant as extracts from different parts of plants can be used as adjuvants to drug treatments [64]. Figueiredo-González [65] investigated the inhibitory activity of α-glucosidase and pancreatic lipase of the crude extract of the native Brazilian fruit cambuí (*Myrcia hatschbachii* D. Legr), commonly found in the Atlantic Forest, which belongs to the same species and family as gabiroba. Similarly, they demonstrated the absence of preliminary toxicity by testing it on the microcrustacean *A. salina*.

Studies have indicated the low toxicity of different parts of gabiroba [66,67,68,69]. In Mato Grosso do Sul, Salmazzo et al. [14] conducted an antiproliferative test and showed that there was no cytotoxicity in the fruit. Its extracts were tested on normal cell lines, in addition to showing promising results against cancer cell lines [18,29].

### 3.5. Seed Germination and Phytotoxicity of WGF Extract

The GF extract initiated germination in the first few days at concentrations of 100 μg/mL (5.66 ± 1.69 μg/mL), 250 μg/mL (11.00 ± 3.74 μg/mL), 500 μg/mL (11.00 ± 0.81), 750 μg/mL (9.66 ± 3, 68 μg/mL), and 1000 μg/mL (8.00 ± 2.44 μg/mL), equivalent to the number of germinated seeds in the water (7.33 ± 2.62 μg/mL) and methanol (9.33 ± 1.24 μg/mL) control groups following the root protrusion count (Figure 4).

The lowest concentration of WGF extract tested (100 μg/mL) exhibited 98.3% seed germination, and the highest concentration tested (1000 μg/mL) exhibited 100% seed germination [70], demonstrating the absence of toxicity.

The extracts were active after seed germination, and the effect of the germination speed on the concentration of the WGF extracts was statistically similar at various concentrations (*p* < 0.0001). The control groups showed a reduced germination speed index (GSI) as a function of the increase in the extract concentration, indicating that germination was not inhibited by the gabiroba extracts. Furthermore, the methanol used as a solvent for the extracts did not prevent the germination process at the various concentrations of GF compared with the methanol control [71] (Figure 5).

The extracts were active after seed germination, and the effect of the germination speed on the concentration of the WGF extracts was statistically similar at various concentrations (*p* < 0.0001). The control groups showed a reduction in the GSI with an increase in the concentration of the extracts, indicating that germination was not inhibited by the gabiroba extracts. Lettuce seeds are known for their high sensitivity, and this may explain why they reacted strongly to the treatments. In addition, we observed that the methanol used as a solvent for the extracts did not prevent the germination process in the various concentrations of WGF in relation to the methanol control [72] (Figure 5).

The various concentrations of the extracts influenced the growth of the hypocotyl of *L. sativa* in a dose-dependent manner (2.95 ± 0.59–5.50 ± 0.65 mm) (*p* < 0.05) (Figure 6). However, no concentration canceled the growth of the aerial parts of lettuce seedlings.

The highest concentration of WGF extract (1000 μg/mL) did not exhibit a positive influence on hypocotyl growth as significantly as the extract at the lowest dose (100 μg/mL). The inhibition mechanism can be explained by the fact that lettuce seeds show variations in the activity of the enzyme endo-β-mannanase when subjected to various concentrations of the extracts. Lower activity of this enzyme, which is considered a determinant in the process of seed germination, results in the weakening of the cell walls of the endosperm [73].

In addition, a reduction in the activity of peroxidase was observed in lettuce seeds subjected to higher concentrations of plant extracts. Notably, peroxidase promotes plant growth, and it directly modulates the development of the hypocotyl, thus elongating the tissue [74].

The allelochemical substances in the extract may inhibit peroxidase activity and consequently reduce growth, especially in the radicle [73], as observed via the germination bioassay. Figure 7 shows that the doses of the WGF 500 (100 μg/mL) and 750 (100 μg/mL) were significant compared with the control group and the WGF 100 μg/mL extract at a lower concentration. The opposite was observed in the WGF 1000 (100 μg/mL) extract group. All concentrations of WGF extracts exhibited a positive influence on radicle length, without significant differences between the concentrations.

Other studies have identified another factor that directly influences this event. Souza et al. [75] attributed the pH concentration at which values of <3.0 and >9.0 can negatively influence the resumption of embryonic growth of *L. sativa*. Such conditions, which are associated with the variation between the osmotic potential of the extracts and the pH, are the primary recommendations described by Rice [76]. In this case, the pH of the WGF was at the lower limit of the suggested recommendations (3.49 ± 0.02) (Table 1). That is, the influence of these factors on germination and initial growth in lettuce seedlings was excluded, reinforcing the hypothesis of an allelopathic effect of WGF extracts on the resumption of lettuce embryonic growth.

In addition, Radouane and Rhim [77] evaluated millet extract (*Pennisetum glaucum*) to reduce the germination of the seeds of cereal species (wheat, oats, and barley). The authors attributed this to the presence of phenolic compounds in the millet extract that can alter the germination process. In this context, WGF at various concentrations did not change the germination process or nullify hypocotyl and radicle growth.

A positive allelopathic effect of the WGF extract on seed germination, GSI, and seedling growth was observed. A similar result was observed in a test performed on two other species of *Campomanesia* cultivated in southern Brazil. Silva et al. [78] evaluated the effects of ethanol extracts from the peels of the fruits of *Campomanesia sessiliflora* O. Berg and *Campomanesia guazumifolia* (Cambess) O. Berg with regard to the effect on germination and the meristem cells of the roots of *Allium cepa* L. The extract of *C. sessiliflora* showed more significant antiproliferative activity and reduced the root size of *A. cepa* seeds. The two extracts did not induce cell death at the concentrations studied, thereby confirming their allelopathic potential.

Pastori et al. [68] demonstrated the genotoxic and antiproliferative activities of the leaves of gabiroba infusions using an in vivo test of *Allium cepa* L. (Amaryllidaceae) cells. Auharek et al. [79] demonstrated that treatment with the leaves of gabiroba reduced reabsorption sites and increased placental weight and the number of live fetuses, citing therapeutic applications. Silva et al. [66] observed no toxicity of hydroethanolic extracts from the leaves of gabiroba in the heart, lungs, spleen, liver, and kidneys of adult rats. However, its mechanism of action remains unknown.

Barbieri et al. [80] isolated pectin from the pulp of gabiroba, which had a monosaccharide composition of 57.7% homogalacturonans and 42.0% type I rhamnogalacturonans. Subsequently, they evaluated the effect of the crude extracts on glioblastoma cells. These fractions induced cytotoxicity (15.55–37.65%) with a positive impact on reactive oxygen species (ROS) levels in human glioblastoma cells, thus showing the antitumor and allelopathic potential of gabiroba pectin.

An important limitation of this study is that the in vitro tests for toxicity are only preliminary. Thus, additional studies investigating the possible hepatotoxic effects of gabiroba extracts using animal models are required to further explore their use as alternative or adjuvant therapies for the treatment of acute and chronic diseases. 

## 4. Conclusions

Our study demonstrates the potential use of gabiroba byproducts in the form of wholemeal flour. In the food industry, it is used as an ingredient in various products because of its high energy value, high protein content, and antioxidant properties. Considering that the preliminary toxicity tests for WGF extracts demonstrated their safe application and significant nutritional value, their consumption in a routine diet as a source of healthy nutrition for regionally vulnerable populations should be encouraged. 

Furthermore, preliminary evidence of its lack of toxicity may contribute to the design of future experimental studies, including double-randomized clinical trials using formulas with various concentrations of the extracts, emphasizing the metabolic benefits at a therapeutic level.

## Figures and Tables

**Figure 1 foods-13-00123-f001:**
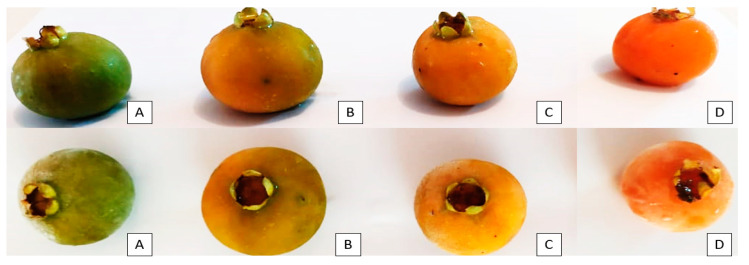
The maturation stages correspond to the minimum and maximum limits of a scale that goes from green fruits (stage 1) (**A**), to intermediate I (stage 2) (**B**), intermediate II (stage 3) (**C**), and ripe (stage 4) (**D**). Fruits grown in the Atlantic Forest from Southern Brazil.

**Figure 2 foods-13-00123-f002:**
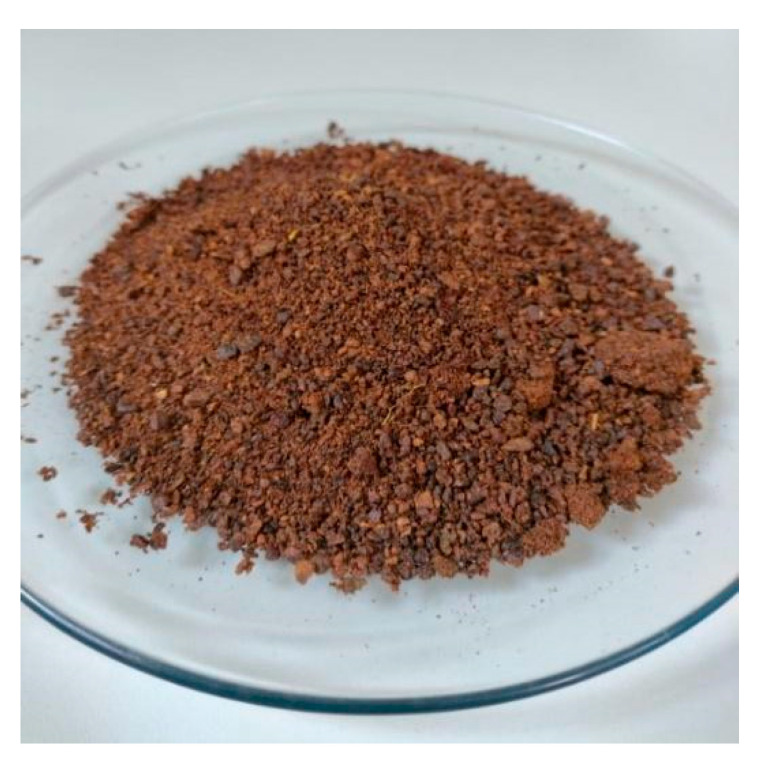
Whole gabiroba flour (WGF).

**Figure 3 foods-13-00123-f003:**
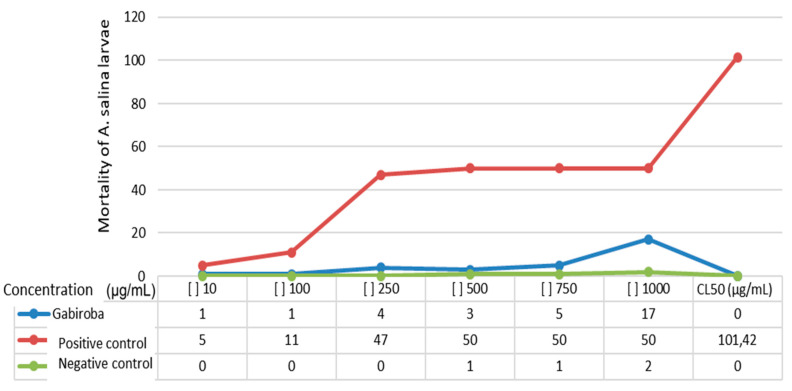
Mortality rate of *Artemia salina* L subjected to different concentrations of GF after 24 h of exposure.

**Figure 4 foods-13-00123-f004:**
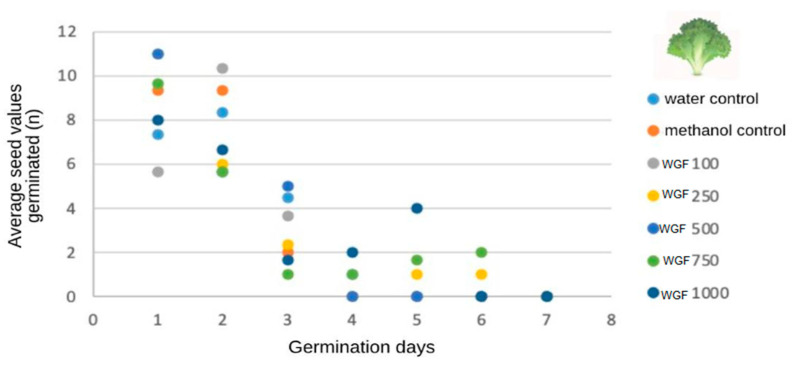
Effect on germination of *Lactuca sativa* seeds treated with controls and whole gabiroba flour extracts.

**Figure 5 foods-13-00123-f005:**
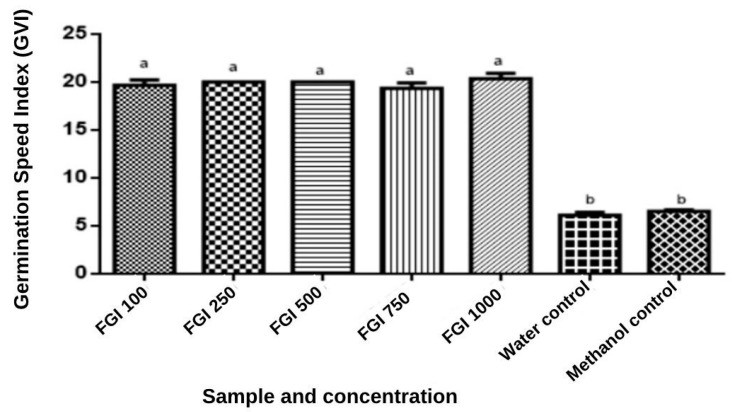
Index of germination speed of lettuce seeds in contact with different gabiroba extracts and controls. Results are expressed as means (mm) and standard deviation in bar graphs. Equal letters show statistical equality using ANOVA followed by Tukey test (*p* < 0.05).

**Figure 6 foods-13-00123-f006:**
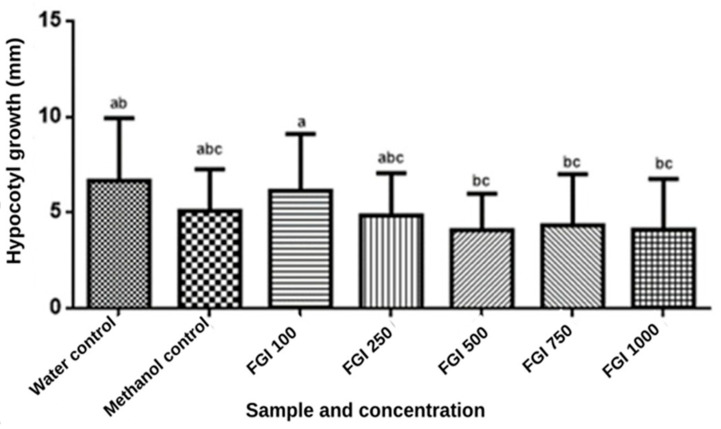
Influence of the length of *Lactuca sativa* hypocotyl in contact with different concentrations of whole gabiroba flour extracts. Results are expressed as means (mm) and standard deviation in bar graphs. Equal letters indicate statistical equality using ANOVA followed by Tukey test (*p* < 0.05).

**Figure 7 foods-13-00123-f007:**
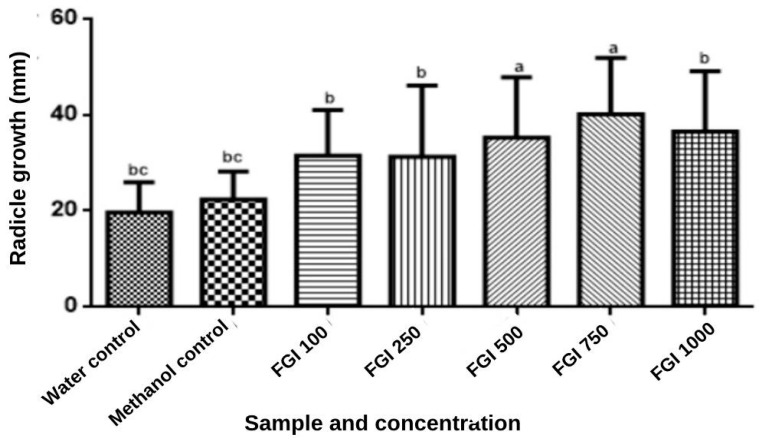
Influence of radicle length of *Lactuca sativa* in contact with different concentrations of whole gabiroba flour extracts. Results are expressed as means (mm) and standard deviation in bar graphs. Equal letters indicate statistical equality using ANOVA followed by Tukey test (*p* < 0.05).

**Table 1 foods-13-00123-t001:** Physicochemical compositions of GF fruit and WGF.

	Gabiroba Fruit (GF) ^1^	Whole Gabiroba Flour (WGF) ^2^
Moisture (g/100 g)	81.18 ± 0.39 ^a^	12.33 ± 0.21 ^b^
Ashes (g/100 g)	0.29 ± 0.07 ^a^	3.03 ± 0.15 ^b^
pH	3.72 ± 0.02 ^a^	3.49 ± 0.02 ^b^
Protein (g/100 g)	0.81 ± 0.64 ^b^	4.75 ± 0.05 ^a^
Lipids (g/100 g)	2.67 ± 0.04 ^b^	8.73 ± 0.52 ^a^
Carbohydrates (g/100 g)	8.63 ± 1.14 ^b^	54.27 ± 0.93 ^a^
Soluble fiber (g/100 g)	1.52 ± 0.74 ^b^	7.15 ± 1.87 ^a^
Insoluble fiber (g/100 g)	4.90 ± 4.37 ^b^	9.74 ± 1.52 ^a^
Total fiber (g/100 g)	6.42 ± 5.11 ^b^	16.89 ± 3.39 ^a^
TEV ^3^ (Kcal)	60.79 ± 13.18 ^b^	314.57 ± 8.66 ^a^

Results obtained with mean values and standard deviation. Different letters on the same line represent statistical differences according to ANOVA statistical test followed by Tukey test. Note ^1^: fresh weight matter (GF); note ^2^: dry weight matter (WGF); note ^3^: TEV (total energy value).

**Table 2 foods-13-00123-t002:** Antioxidant activities of GF and whole WGF using different methods.

Fraction	DPPH (µmol TEAC.100 g^−1^)	ORAC (µmol TEAC.100 g^−1^)	FRAP (µmol TEAC.100 g^−1^)
GF ^1^	380.64 ± 43.62 ^a^	2824.00 ± 11.79	126.97 ± 25.57
WGF ^2^	681.72 ± 12.35 ^b^	NA ^3^	NA ^3^
*p*-value	0.0003	-	-

Note ^1^: GF—gabiroba fruit; note ^2^ WGF—whole gabiroba flour; note ^3^: NA—fraction not analyzed. Results are represented as means ± standard deviation; ^a,b^ different letters in the same column represent statistical difference using the ANOVA statistical test, followed by Tukey test (*p* < 0.05).

## Data Availability

Data is contained within the article.

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
