# Peer review of "Cytotoxic and Phytotoxic Activities of Native Brazilian Forest Gabiroba (Campomanesia xanthocarpa Berg.), Fruits, and Flour against Shrimp (Artemia salina L.) and Lettuce (Lactuca sativa L.)"

_foods, 2023, doi:10.3390/foods13010123_

Round 1

Reviewer 1 Report

Comments and Suggestions for Authors

v General evaluation:  aim, significance, and novelty

The aim of this study was to analyze the physicochemical characterization, antioxidant activity of gabiroba fruit (GF), and whole gabiroba flour (WGF), and the toxicity of its extract.

This should be better restricted to cytotoxicity, and acute phytotoxicity of both gabiroba fruit and flour since the physical-chemical composition and antioxidant activity have been previously handled. However it can still confirm the previous results.

This fruit is not yet known as a common food and is specific to certain region in the Brazilian Atlantic Forest biome. So, this new food may have new added values, specifications, and applications. So, the article should stress that dimension in introducing this new food product and explain in the conclusive part of the discussion how this new food has new added values and novel prospects while stressing the Cytotoxic and phytotoxic activities of it. This should be also confirmed in the conclusions section.

v The title

The title should be expressing the content and highlighting the novelty. It is suggested to be as follows:

Cytotoxic and phytotoxic activities of native Brazilian Forest  gabiroba (Campomanesia xanthocarpa Berg.), fruits and flour against shrimp (Artemia salina L.) and lettuce (Lactuca sativa L.)

v Abstract

·        The abstract should clearly but briefly indicate the novelty and significance of the research-work.

·        The details of methodologies are not required in this section to allow more space for the content and results. So, part L24-29 should be rephrased to remove unnecessary details in this part. It is sufficient to briefly present the conducted analyses. 

·        Presenting the results should include the most important and promising results referring to their significance.

·        The contents of the constituents should be indicated if they are on fresh or dry weight basis.

·        Explain what does it mean 90% yield? % of what.

·        Specify the toxicity. Against what. And the survival of what?

·        Linguistic modifications:

1.      L22, change [of the fruit (GF)] into (of gabiroba fruit (GF)]

2.      L22, change (whole gabiroba flour (WGF), and toxicity of the WGF extract) into (whole gabiroba flour (WGF), and the toxicity of its extract)

3.      L38, change (. We conclude that both the fruit, and WGF) into (. It is conclude that GF, and WGF)

v Introduction

·        After referring  scientific name of gabiroba (Campomanesia 4 xanthocarpa Berg.), please use either name throughout the manuscript solely and not interchangeably. ((Campomanesia was used 19 times while gabiroba 69 times). Please eliminate confusion.

·        The introduction should clearly indicate the novelty of the current work. For example the last paragraph (L82-86) may be rephrased as follows: ((The absence of investigations on the allelopathic activity of gabiroba necessitates its exploration. Thus, the present work principally targeted the cytotoxicity, and acute phytotoxicity of both gabiroba fruit and flour, and portrayed their physical-chemical composition and antioxidant activity.))

·        Linguistic modifications

1.      L45, change (an important role) to (a vital role).

2.      L76, change (which is based on the chemical) to( adopting the chemical)

v Materials and methods

·        Explain clearly how was the gabiroba flour prepared?

·        L116, change (Initially, the aqueous extract of the fruits, and the WGF was obtained) into (Initially, the aqueous extracts of the fruits, and the WGF were obtained)ز

·        Once gabiroba fruits and whole gabiroba flour were abbreviated as (GF) and (WGF), the abbreviations ions should replace the complete names allover the text to avoid confusion. For example, the title (2.1.6.) should read ( Phytotoxicity bioassay of WGF extract ) and the title (2.1.6.2.) should read (Growth test of WFG on seedlings)

v Illustrations

o   Under Table 2 indicate clearly when the determination were on fresh weight matter (GF) and dry weight matter (WGF).

o   Modify the title of Table 3 to: Table 3. Antioxidant activity of gabiroba fruits (GF), and whole gabiroba flour (WGF) by different method. Remove the phrase (WGF: whole gabiroba flour) under the table.

o   Figure 1. Explain what do the letters A, B, C and D refer to.

o   Figure 3, complete the title to be more indicative: Figure 3. Mortality rate of Artemia salina L subjected to different concentrations of GF after 24 h of exposure.

o   Figure 4, complete the title (Figure 4. Effect different WGF concentrations on the germination of Lactuca sativa seeds treated. The data on the Figure should be in English. (They are in Portuguese).

o   The data on Figures 5, 6 and 7 are written in Portuguese and not in English. This should be duly corrected. The titles must be reformulated and corrected to reflect the content.

v Results and discussion

General

This section should illustrate how could the study reveal, evaluate and exploit the Cytotoxic and phytotoxic activities of these new plant products. The discussion should treat this as the main objective of the study.

The section (3.4. Acute in vitro cytotoxicity of whole gabiroba flour extract) should be presented in more detail while the section 3.2. (Physicochemical characterization of the whole gabiroba fruit and flour) should be abbreviated to put more stress on the second topic.

Minor modifications

L221, change (allowing for an important yield) into (allowing an essential yield)

L227, change (Results expressed) into (Results are expressed)

L232, change (phenotypic characteristics, that are determined according to) into (phenotypic characteristics determined by)

L252-253, change (, in accordance with the legislation's recommendations) into (, conforming to the legislation's recommendations,)

L367, change into (3.4. Acute in vitro cytotoxicity of WGF extract)

L378, change (, allowing it to be applied for dietary purposes) into (, enabling its dietary applications).

L393-394, change (be positively correlated with certain biological) into (can be associated with specific biological activities).

L439, change (The WGF 1000 extract at its highest concentration (1000 µg/mL) into (The highest concentration of WGF extract (1000 µg/mL)

L452, change (and as a result reducing growth,) into (and consequently reducing growth,)

L456-458, change into (All the WGF extracts showed a positive influence on radicle length in the different concentrations without significant differences between the levels of concentrations.).

v Conclusions

This part is nearly not presenting the main objective of the research while it highly should. So please explain in clear statement how did the study revealed the Cytotoxic and phytotoxic activities and their relation to the real value of this plant byproducts.

Comments on the Quality of English Language

The article needs extensive English editing

Author Response

Manuscript ID Foods-2708043

Dear Editor,
Corrections made to the article are highlighted in blue in the text.
Please find attached the revised version of the manuscript and the letter responding to the suggestions.

Kind regards,

Aiane Benevide Sereno.

Reviewer 2 Report

Comments and Suggestions for Authors

The aim of the presented work is to evaluate physicochemical composition and antioxidant activity of gabiroba and its stabilized flour, and also to analyze the cytotoxicity and phytotoxicity of the whole gabiroba flour extract. The topic is interesting, the paper is written appropriately, but I advise the authors to take the following comments into account:

Line 89, Specify the time period and collection season (year, month).

Line 103, Producer, city, country for Freeze Dryer

Line 256, Is not Fernandes et al.? Also, this reference in the text is marked with the number 55 and in the reference section under the number 51 (Line 79, Molish in the text 25 in the References 22) the same applies to all the others. Check all references and correct.

Line 261-292, Compare the obtained mineral and protein content in the fresh fruit and whole gabiroba flour with the mineral and protein content obtained by other authors for gabiroba or similar fruits.

Line 284, Why were levels of total solids for WGF not analyzed?

Line 301, fruit. [65]. The dot is redundant.

Line 344-345, Although statistical differences between GF, and WGF were not demonstrated (p = 0.0003) by this method? How not? Different letters represent statistical difference.

Table 3, Why was the determination of the antioxidant activity of WGF not done using the ORAC and FRAP methods? It would be useful to confirm that the antioxidant activity of flour does not decrease and for comparison.

Line 266-267, .. x 01 kg dry, Do you mean 1 kg? Is it an error? Please, clarify and correct it.

Line 382, Figueiredo-González [87] et al. is missing.

Line 400-404, This paragraph is redundant. Remove it.

Figure 4, 5, 6 7. The language in the figures is not English.

The conclusion is short and inconspicuous. Include future studies, limitations, and highlight practical applications.

Author Response

(The authors gave the same response as above.)

Round 2

Reviewer 1 Report

Comments and Suggestions for Authors

The manuscript has been dedicatedly revised and improved accordingly except of few minor required modifications as follows:

·        Change the first sentence in the abstract into (Gabiroba, a native fruit in Brazil's Atlantic Forest region, has significant nutritional and therapeutic properties.)

·        L27-28, modify into (In the present study, physicochemical analyses of fresh fruits (GF), dehydrated whole gabiroba……..)

·        L117, change into (According to the manufacturer's information)

·        L124, change into (through the forced hot air circulation.)

·        Modify slightly the title of Table 2 into (Table 2.  Antioxidant activity of GF, and whole WGF by different methods.)

·        The authors did not explain what do the letters A, B, C and D refer to in Figure 1. Under the stable they refer to four stages of maturation by stage 1, stage 2, stage 3 and stage 4 while in the figure they are indicated by A, B, C and D. Please unify the presentation mode or just link of ways of referencing.

Comments on the Quality of English Language

English quality has been improved

Author Response

Manuscript ID Foods-2708043

Dear reviewer,

Corrections made to the article are highlighted in blue in the text.

Please find attached the revised version of the manuscript. Below are responses to reviewers.

The manuscript has been dedicatedly revised and improved accordingly except of few minor required modifications as follows:

  • Change the first sentence in the abstract into (Gabiroba, a native fruit in Brazil's Atlantic Forest region, has significant nutritional and therapeutic properties.)
  • L27-28, modify into (In the present study, physicochemical analyses of fresh fruits (GF), dehydrated whole gabiroba……..)

R: Change made.

  • L117, change into (According to the manufacturer's information)

R: Change made.

  • L124, change into (through the forced hot air circulation.)

R: Change made.

  • Modify slightly the title of Table 2 into (Table 2.  Antioxidant activity of GF, and whole WGF by different methods.)

R: Change made.

  • The authors did not explain what do the letters A, B, C and D refer to in Figure 1. Under the stable they refer to four stages of maturation by stage 1, stage 2, stage 3 and stage 4 while in the figure they are indicated by A, B, C and D. Please unify the presentation mode or just link of ways of referencing.

R: Change made.

King Regards,

Aiane Benevide Sereno.
